# Manganese Porphyrin-Based SOD Mimetics Produce Polysulfides from Hydrogen Sulfide

**DOI:** 10.3390/antiox8120639

**Published:** 2019-12-12

**Authors:** Kenneth R. Olson, Yan Gao, Faihaan Arif, Shivali Patel, Xiaotong Yuan, Varun Mannam, Scott Howard, Ines Batinic-Haberle, Jon Fukuto, Magdalena Minnion, Martin Feelisch, Karl D. Straub

**Affiliations:** 1Indiana University School of Medicine—South Bend, South Bend, IN 46617, USA; yangao@iu.edu (Y.G.); farif1234@gmail.com (F.A.); spatel01@saintmarys.edu (S.P.); 2Department of Biological Sciences, University of Notre Dame, Notre Dame, IN 46556, USA; 3Department of Electrical Engineering, University of Notre Dame, Notre Dame, IN 46556, USA; xyuan2@nd.edu (X.Y.); vmannam@nd.edu (V.M.); showard@nd.edu (S.H.); 4Department of Radiation Oncology, School of Medicine, Duke University, Durham, NC 27710, USA; ibatinic@duke.edu; 5Department of Chemistry, Sonoma State University, Rohnert Park, CA 94928, USA; fukuto@sonoma.edu; 6NIHR Southampton Biomedical Research Center, University of Southampton, Southampton, General Hospital, Southampton SO16 6YD, UK; M.Minnion@soton.ac.uk (M.M.); M.Feelisch@soton.ac.uk (M.F.); 7Clinical & Experimental Sciences, Faculty of Medicine, Southampton General Hospital and Institute for Life Sciences, University of Southampton, Southampton SO16 6YD, UK; 8Central Arkansas Veteran’s Healthcare System, Little Rock, AR 72205, USA; Karl.Straub@va.gov; 9Departments of Medicine and Biochemistry, University of Arkansas for Medical Sciences, Little Rock, AR 72202, USA

**Keywords:** H_2_S, reactive sulfur species, reactive oxygen species, Mn porphyrins, BMX-001, antioxidants

## Abstract

Manganese-centered porphyrins (MnPs), MnTE-2-PyP^5+^ (MnTE), MnTnHex-2-PyP^5+^ (MnTnHex), and MnTnBuOE-2-PyP^5+^ (MnTnBuOE) have received considerable attention because of their ability to serve as superoxide dismutase (SOD) mimetics thereby producing hydrogen peroxide (H_2_O_2_), and oxidants of ascorbate and simple aminothiols or protein thiols. MnTE-2-PyP^5+^ and MnTnBuOE-2-PyP^5+^ are now in five Phase II clinical trials warranting further exploration of their rich redox-based biology. Previously, we reported that SOD is also a sulfide oxidase catalyzing the oxidation of hydrogen sulfide (H_2_S) to hydrogen persulfide (H_2_S_2_) and longer-chain polysulfides (H_2_S_n_, *n* = 3–7). We hypothesized that MnPs may have similar actions on sulfide metabolism. H_2_S and polysulfides were monitored in fluorimetric assays with 7-azido-4-methylcoumarin (AzMC) and 3′,6′-di(O-thiosalicyl)fluorescein (SSP4), respectively, and specific polysulfides were further identified by mass spectrometry. MnPs concentration-dependently consumed H_2_S and produced H_2_S_2_ and subsequently longer-chain polysulfides. This reaction appeared to be O_2_-dependent. MnP absorbance spectra exhibited wavelength shifts in the Soret and Q bands characteristic of sulfide-mediated reduction of Mn. Taken together, our results suggest that MnPs can become efficacious activators of a variety of cytoprotective processes by acting as sulfide oxidation catalysts generating per/polysulfides.

## 1. Introduction

The key component of many enzymes and respiratory pigments is a porphyrin ring that stabilizes a reactive metal, often iron, in its center. These iron porphyrins can catalyze a variety of reduction/oxidation (redox) reactions or they can non-catalytically coordinate with and transport small gaseous molecules especially oxygen e.g., in hemoglobin, myoglobin, and neuroglobin [1,2,3,4,5,6,7,8]. The substitution of the metal center can substantially affect the catalytic properties of the metal, and this can be further “fine-tuned” by modifying the specific porphyrin in which it is contained.

Manganese-centered porphyrins (MnPs) have received considerable attention in this regard as manganese readily undergoes redox reactions and the porphyrin ring can be modified to achieve the specific redox potentials of the manganese [9,10,11]. Most notable are three MnPs; MnTE-2-PyP^5+^ (MnTE, AEOL10113, BMX-010), MnTnHex-2-PyP^5+^ (MnHex), and MnTnBuOE-2-PyP^5+^ (MnBuOE, BMX-001), that closely mimic the reduction potential (*E_½_*) of the endogenous antioxidant enzyme, superoxide dismutase (SOD; ~+300 mV vs the normal hydrogen electrode (NHE)) [10,11]. Although these MnPs are excellent SOD mimetics, they lack the tertiary structure that enables the high specificity of SOD toward superoxide and they react with other compounds such as peroxynitrite, carbonate radical, nitric oxide, peroxide, small aminothiols or protein thiols, and hypochlorite [12,13].

These SOD-mimetic properties of MnPs notwithstanding, many of the biological effects of MnPs have been attributed to their ability to generate hydrogen peroxide (H_2_O_2_). In this reaction the Mn, which is introduced as Mn(III), is reduced by an intracellular reductant such as ascorbate to Mn(II). It is then reoxidized by O_2_, which in most circumstances is presumably more abundant in cells than superoxide, to Mn(III), forming superoxide in the process. As this cycle is repeated, the superoxide that is generated either spontaneously, or catalyzed by SOD or MnPs, is dismuted to O_2_ and H_2_O_2_. The resultant H_2_O_2_ is used by MnP in a catalytic cycle where regulatory protein cysteines, that may mediate cytoprotection in healthy cells and/or initiate apoptotic reactions in malignant ones, are oxidized [12,13].

There are a number of factors that suggest MnPs may also react with sulfide and reactive sulfur species (RSS) in addition to reacting with O_2_ and other reactive oxygen species (ROS) and reactive nitrogen species (RNS). First, ROS generated by one-electron reductions of O_2_, (O_2_•^−^, H_2_O_2_, and •OH) are chemically similar to RSS generated by sequential one-electron oxidations of hydrogen sulfide (H_2_S). These include the thiyl radical (•SH), persufide (H_2_S_2_), persulfide radical, i.e., “supersulfide” (S_2_•^−^) and finally elemental sulfur (S_2_); the latter may ultimately cyclize to S_8_ [14]. Second, the downstream effectors activated by ROS, mainly cysteine sulfur on regulatory proteins, are essentially the same as those activated by RSS [15,16,17,18,19,20,21,22,23]. Third, not only are ROS and RSS chemically and biologically similar, but five of the most common methods used to measure ROS also detect RSS, with greater or equal sensitivity suggesting that even distinguishing between ROS and RSS in cells is challenging [24]. Fourth, we have recently shown that both Cu/Zn- and MnSOD oxidize H_2_S and form polysulfides [25]. Fifth, MnPs catalyze a one-electron oxidization of either cysteine or glutathione to form cystine or GSSG, respectively [12]. Collectively, these observations provide a strong rationale for potential reactions of MnPs with inorganic RSS.

The present study examines the reactions of MnPs with hydrogen sulfide (H_2_S). We show that MnTE, MnTnHex, and MnTnBuOE react with H_2_S to produce H_2_S_2_ and subsequently longer-chain polysulfides, and this process consumes O_2_ in the process. Given the uncertainty in distinguishing between ROS and RSS, our results suggest that some, perhaps even a substantial portion, of the biological effects of MnPs ascribed to ROS metabolism may, in fact, be due to RSS metabolism.

## 2. Materials and Methods

### 2.1. Chemicals

SSP4 (3′,6′-di(*O*-thiosalicyl)fluorescein), Na_2_S_2_, Na_2_S_3_, and Na_2_S_4_ were purchased from Dojindo molecular Technologies Inc. (Rockville, MD, USA). MnTBAP, which is also a Mn porphyrin, but lacks appreciable SOD-mimetic activity [26] was purchased from Sigma–Aldrich (St. Louis, MO, USA), other MnPs were synthesized as previously reported [27,28]. All other chemicals were purchased either from Sigma-Aldrich or ThermoFisher Scientific (Grand Island, NY, USA). Please note that we use H_2_S to denote the total sulfide added (sum of H_2_S + HS^−^) usually derived from Na_2_S. Also, while S^2−^ is often thought as part of the H_2_S + HS^−^ equilibrium, it does not exist under these conditions as the pK > 12 [29]. Phosphate buffer (PBS; in mM): 137 NaCl, 2.7 KCl, 8 Na_2_HPO_4_, 2 NaH_2_PO_4_. pH was adjusted with 10 mM HCl or NaOH to pH 7.4.

### 2.2. Fluorescence Measurements

Compounds of interest were aliquoted into black 96-well plates in a darkened room and fluorescence was measured on a SpectraMax M5e plate reader (Molecular Devices, Sunnyvale, CA, USA). Fluorescence was typically measured every 10 min for at least 90 min. Excitation/emission wavelengths for 3′,6′-di(*O*-thiosalicyl)fluorescein (SSP4), and 7-azido-4-methylcoumarin (AzMC) were 482/515 and 365/450 nm, respectively, as per the manufacture’s recommendations.

### 2.3. Hypoxia

Hypoxia experiments were performed in a model 856-HYPO hypoxia chamber (Plas Labs, Inc., Lansing, MI, USA) in either 21% O_2_/79% N_2_ (normoxia) or 100 % N_2_ (hypoxia). The latter reduced O_2_ to <0.35% at room temperature (20 °C). For hypoxia experiments, the reactants were allowed to react in the hypoxia chamber for 90 min and then were covered with the plate cover prior to moving them to the plate reader which was in room air (21% O_2_).

### 2.4. Mass Spectrometry

Ultrahigh performance liquid chromatography tandem mass spectrometry (UPLC-MS/MS) detection was used to identify and quantify polysulfides formed from the reaction of MnTE and MnTnHex with H_2_S (added as Na_2_S). Polysulfide formation at each time point was captured following derivatization by iodoacetamide (IAM) for 30 min at room temperature. Because of the lack of stable authentic reference standards for IAM-derivatized polysulfides (which prevented us from constructing concentration/response curves for individual polysulfides, and determining their ionization efficiency, etc.) no exact concentrations could be calculated; consequently we report results from these experiments in the form of “peak areas.”

The derivatized polysulfide compositions were analyzed using a Waters Aquity UPLC system coupled to a tandem quadrupole mass spectrometer (Xevo TQ-S, Waters, Wilmslow, Cheshire, UK); a mixed mode 1.6 µm Modus 100 × 2.2 mm Aqua UPLC column (Chromatography Direct, Runcorn, Cheshire, UK) kept at 30 °C was used for the separation. Mobile phase A was 5 mM ammonium formate in water with 0.15% formic acid; mobile phase B was 5 mM ammonium formate in 95% acetonitrile with 5% H_2_O and 0.15% formic acid. The gradient was as follows: 99% A decreasing to 60% A over 4.5 min, then down to 0% A over 0.5 min and maintained at that level for 1.5 min. The column was then equilibrated back to 99% A over 0.5 min and maintained at 99% A for an additional 1 min. The flow rate was 0.2 mL/min and the injection volume was 5 µL. Mass spectrometry settings are as follows: capillary voltage 3.0 kV, source offset 5 V, desolvation gas flow 800 L/h, cone gas flow 150 L/h, nebulizer pressure 7.0 bar, collision gas flow 0.14 mL/min, desolvation temperature 400 °C. The following MRM transitions were used for the detection of IAM-derivatized sulfide and polysulfide species: 149 > 104 (IAM_2_-S_1_), 181 > 91 (IAM_2_-S_2_), 213 > 91 (IAM_2_-S_3_), 245 > 91 (IAM_2_-S4), 277 > 91 (IAM_2_-S_5_), 309 > 91 (IAM_2_-S_6_), and 341 > 91 (IAM_2_-S_7_). Cone and collision energies were 8 V and 12 V, respectively.

### 2.5. Absorbance Spectroscopy

Absorbance and difference spectra were examined on the Spectramax M5e plate reader (Molecular Devices, Sunnyvale, CA, USA). In a typical experiment the MnP of interest was added and absorbance measured by scanned at 1 nm/s, at 1 nm intervals. H_2_S or H_2_S_2_ was then added and absorbance was measured again at varying intervals. Difference spectra were obtained by subtracting the MnP absorbances from the H_2_S-treated absorbances. A minimum of three spectra were obtained for each experiment, figures 8–10 show a typical trace.

### 2.6. Oxygen Consumption

Oxygen was measured with a FireStingO2 oxygen-sensing system (Pyroscience Sensor Technology, Aachen, Germany) using a non-oxygen consuming 3 mm dia OXROB10 fiberoptic probe. The system was calibrated in room air and under a stream of nitrogen. Buffer (PBS) was sparged with nitrogen for 20 min to reduce the O_2_ and placed in a 5 mL glass vial, the probe was inserted into the buffer and sealed with parafilm^®^ (SigmAldrich, St. Louis, MO, USA). The volume of buffer was sufficient to eliminate nearly all head space. Stabilization, H_2_S (Na_2_S) was injected through the parafilm via a microliter syringe to produce a final concentration of 300 μM and the system was resealed and again allowed to stabilize. MnTnBuOE (10 μM final concentration) was then injected and O_2_ measured until readings became stable. The order of H_2_S and MnTnBuOE addition was reversed in additional experiments to verify that both H_2_S and the MnP were required to consume O_2_.

### 2.7. Calculations

Results are expressed as mean +SE. Statistical analysis was determined by one-way ANOVA with Holm-Sidak for multiple comparisons. Significance was assumed at *p* ≤ 0.05.

## 3. Results

### 3.1. MnP-Catalyzed Polysulfide Generation from H_2_S with Variable MnP in PBS

In initial studies we determined if MnTE and MnTnBuOE concentration-dependently produced polysulfides from H_2_S (added as Na_2_S) by measuring SSP4 fluorescence over 90 min. Both MnTE and MnTnBuOE increased SSP4 fluorescence in the presence of 100 μM H_2_S from 0.1 to 1 μM MnP, but concentration-dependently decreased the fluorescence thereafter (Figure 1). 

MnPs had minimal effects on SSP4 fluorescence in the absence of H_2_S (compare ordinate scales). These results suggest that both MnTE and MnTnBuOE generate polysulfides at low concentrations but inhibit fluorescence at higher concentrations.

We previously showed that porphyrins and porphyrin-containing proteins optically interfere with excitation/emission of fluorophores at lower wavelengths (~300–550 nm) [30]. To determine if the effects of MnPs were on polysulfide production or due to physical (optical) interference with fluorescence itself, we compared the effects of MnPs on fluorescence produced by 1 μM fluorescein at 90 min to the 90 min SSP4 fluorescence samples of MnPs reacting with 100 μM H_2_S. Both MnPs concentration-dependently inhibited fluorescence from both SSP4 and fluorescein at 10 μM MnP, indicative of nonspecific optical quenching (Figure 1C,F).

The concentration- and time-dependent effects of MnTE, MnTnHex, and MnTnBuOE were then examined over smaller increments of MnP (Figure 2). All MnPs concentration-dependently increased polysulfide production (SSP4 fluorescence) at low MnP concentrations and decreased fluorescence at high concentrations, the latter consistent with optical interference (Figure 1). MnTE was the most efficacious and MnTnHex the least efficacious compound. Maximum SSP4 fluorescence at 100 μM H_2_S was achieved with 0.3 μM MnTE. More than 50% of maximum SSP4 fluorescence was produced with 0.1 μM MnTE. Because of the apparent optical interference of MnPs with SSP4 fluorescence, MnP concentrations were limited to 3 μM in further fluorometric studies.

### 3.2. MnP-Catalyzed Polysulfide Generation from Variable H_2_S in PBS

Concentration- and time-dependent polysulfide production (SSP4 fluorescence) from H_2_S by MnTBAP, MnTE, MnHex, and MnTnBuOE (all 3 μM) are shown in Figure 3A–E and compared to SSP4 fluorescence from a variety of polysulfide standards (Figure 3F). Compared to the other MnPs, MnTBAP was minimally efficacious. All other MnPs concentration- and time-dependently increased polysulfide concentration. MnTE was again the most efficacious and all reactions appeared to be completed, or nearly so, by 90 min.

To further evaluate a possible reaction between MnPs and polysulfides we incubated 1 μM MnTE with 300 μM of the mixed polysulfide, K_2_S_n_ (*n* = 1–7, where the ratio of sulfur species is unknown) or polysulfide standards, Na_2_S_2_ or Na_2_S_4_, for either 0, 30, 60, or 90 min before adding SSP4. As shown in Figure 3G, when SSP4 was added 30 min after MnTE and K_2_S_n_ the fluorescence was decreased by nearly forty percent of that observed when SSP4 was added immediately suggesting that MnTE reacts with one or more of the sulfur moieties in K_2_S_n_. Allowing additional time for the reaction to proceed before adding SSP4 did not result in any additional decrease in SSP4 fluorescence indicating that the reaction between MnTE and the sulfur species was completed in 30 min. This profile was somewhat different when MnTE was incubated with either Na_2_S_2_ or Na_2_S_4_, taking 90 min to fall to the same level of fluorescence as K_2_S_n_ at 30 min. As a fraction of K_2_S_n_ is H_2_S, volatile loss of H_2_S could account for this difference. Nevertheless, these results suggest that MnTE may react with polysulfides. It is not known why the time course for this is much longer than that observed with the mass spectrometric measurements described in the following section, although the relatively slow kinetics of the SSP4-polysulfide may have been a contributing factor.

### 3.3. Mass Spectrometric Identification of Polysulfides Produced from H_2_S and MnPs

In preliminary experiments we examined polysulfides produced from the reaction of 1 μM MnTnHex with 100, 300, and 1000 μM H_2_S (added as Na_2_S). With 1 μM MnTnHex we could detect polysulfides only with Na_2_S concentrations >100 μM. At 300 μM Na_2_S with all three MnPs we could detect both H_2_S and H_2_S_2_ and with 1 mM Na_2_S we could detect H_2_S, as well as polysulfides from H_2_S_2_ through H_2_S_5_ (not shown). One mM Na_2_S was therefore used in all subsequent experiments.

The time course of the reactions of 1 mM Na_2_S with 1 μM MnTE and 1 μM MnTnHex are shown in Figure 4A–C and D–F, respectively. MnTE produced a rapid, exponential decrease in H_2_S with an apparent rate constant of 0.051 min^−1^ and half-time of 13.7 min. H_2_S_2_ was the predominant polysulfide produced peaking at approximately 20 min and declining thereafter with an apparent exponential rate constant of 0.050 min^−1^ (t ½ =14.0 min), similar to the rate of consumption of H_2_S. H_2_S_3_ reached a maximum around 30 min followed by lesser amounts of H_2_S_5_ and H_2_S_4_, both peaking at around 50 min. There was also a small amount of H_2_S_6_ produced. H_2_S reaction with MnTnHex appeared slightly slower, most likely because of steric constraints as seen in the reaction of MnTE vs. MnTnHex with ascorbate [31]. After a brief delay, H_2_S concentrations began an exponential decrease with a longer rate constant (0.043 min^−1^; t ½ = 16.2 min) than that for MnTE. With MnTnHex H_2_S_2_ peaked at 30 min and the rate of decline was somewhat slower (0.035 min^−1^; t ½ = 20.1 min). However, the peak areas for H_2_S_2_ were similar for the two MnPs because of their very similar thermodynamic properties [32]. H_2_S_3_ peaked at 50 min followed by H_2_S_5_ and H_2_S_4_ at around 1 h. MnTnHex appeared to produce more polysulfides (in particular, S_3_ and S_6_) than MnTE and they generally persisted longer in solution, although as stated above, the lack of appropriate calibration standards precluded definitive quantification.

We next evaluated the effect of MnTE concentration on the rate of H_2_S consumption and polysulfide production. As shown in Figure 5, the rate of H_2_S consumption increased as the concentration of MnTE increased. The rates of polysulfide production and disappearance also increased with increasing MnTE concentration. The peak concentration of most polysulfides, especially H_2_S_4_ and above, also increased with MnTE concentration. The peak concentration of H_2_S_2_ was the lowest with 10 μM MnTE. Most likely this was due to a more rapid conversion to longer-chain polysulfides. These results confirm our observation that increasing the concentration of MnPs increases the rate of H_2_S metabolism and polysulfide formation and turnover as shown in Figure 2.

To determine if MnTE reacts with the IAM-derivatized polysulfides we incubated 1 mM Na_2_S_4_ with 100 mM IAM for 30 min before adding various concentrations of MnTE to aliquots of this mixture. As shown in Figure 6, the peak area for H_2_S and H_2_S_2_ decreased as MnTE concentration increased, H_2_S_3_ was unaffected, and the areas for H_2_S_4_, H_2_S_5_, and H_2_S_6_ all increased. These results suggest that MnTE reacts to some extent even with derivatized H_2_S and H_2_S_2_ to form longer-chain sulfur derivatives. Thus, some of the longer-chain polysulfides detected in the reaction between MnPs and H_2_S may originate from this side reaction.

### 3.4. Effects of Oxygen on MnP Catalyzed Polysulfide Production

We previously observed that polysulfide production from SOD-derived metabolism of H_2_S was oxygen dependent [25]. To determine if oxygen was also required for MnP reactions with H_2_S we measured polysulfide production from 300 μM H_2_S in both normoxia and hypoxia. As shown in Figure 7A, polysulfide production by all three MnPs after 90 min in hypoxia (t = 0 min in normoxia), was somewhat greater than polysulfide production after 90 min in normoxia. These results suggest that MnP-catalyzed oxidation of H_2_S is oxygen independent. Polysulfide production continued to increase in the hypoxic samples for the ensuing 90 min after they were removed from the hypoxia chamber and exceeded that of the 90 min normoxic samples.

However, although our hypoxia chamber reduced the O_2_ level to ~0.0035% (~5 μM), this did not completely eliminate the possibility that oxygen was involved in the reaction especially given the large volume of air inside the hypoxia chamber. To examine the possibility that O_2_ was involved, we monitored the oxygen consumption with a non-oxygen consuming probe during the reaction of MnTnBuOE with H_2_S. Figure 7B–D shows that oxygen was indeed consumed and that this required both H_2_S and the MnP. These experiments not only show that O_2_ is consumed, but they also show that MnTnBuOE serves as a catalyst because the initial MnTnBuOE concentration (1 μM) was far lower than the amount of O_2_ consumed (70 and 175 μM in Figure 7C,D, respectively). The apparent inhibitory effect seen in samples incubated in room air compared to those incubated in the hypoxia chamber (Figure 7A) suggests that some of the sulfur produced was a sulfur oxide that was not detected by SSP4.

### 3.5. MnP Absorbance Spectra

Although the most common oxidation states of manganese range from +2 to +7, in the MnPs they range from +2 to +5 Mn(II) to Mn(IV) [33]. The hallmark of MnP-catalyzed H_2_O_2_ production from a reductant such as ascorbate and O_2_ is the one-electron redox cycling of manganese between Mn(III)P and Mn(II)P [13], which is favored over a wide range of pH from ~3 to 10 [34]. In this one-electron redox reaction the Soret peak blue-shifts from 454 to 438 nm and the Q band red-shifts from 560 to 565 nm as the MnP is reduced from Mn(III)P to Mn(II)P [35]. In the present studies we compared these spectral changes to measurements of polysulfide production to gain further insight into the mechanism of MnP-driven catalysis of polysulfide production from H_2_S.

#### 3.5.1. Effects of H_2_S on MnP Oxidation State

In order to determine if H_2_S affects the manganese oxidation state, we added 100 μM H_2_S to 1 μM MnPs and monitored absorbance between 350 and 650 nm wavelength at 5 min intervals. The Soret band of MnTE was 452 mn and after addition of H_2_S it shifted to 440 nm at t = 0 min, with a slight red shoulder, then to 437 nm at 10 min and stabilized at 436 nm by 15 min (Figure 8A,D). The Q band (Figure 8A, inset) shifted from 557 nm prior to H_2_S to 562 nm and remained there. These results suggest that H_2_S reduced Mn from Mn(III) to Mn(II). The Soret band peak of MnTnHex was also 452 nm prior to H_2_S and underwent a blue shift to 439 nm after H_2_S and the Soret band of MnTnBuOE went from 453 nm to 440 nm after H_2_S suggesting a similar reduction of Mn (not shown). Because of essentially identical thermodynamics of three cationic MnPs (controlling the types and amount of polysulfide products) the rest of the studies were carried out with MnTE; MnTnHex and MnTnBuOE were not examined further.

#### 3.5.2. Effects of DTT and H_2_O_2_ on MnTE and MnTE/H_2_S Spectrum

To confirm that Mn was reduced by H_2_S we then monitored the MnTE spectrum for 20 min after addition of 1 mM DTT and then for another 20 min after addition of 100 μM H_2_S. DTT produced a blue-shift in the Soret band from 451 nm to 438 nm, again with a slight right shoulder at t = 0 min stabilizing at 437 nm thereafter (Figure 8B,E). Subsequent addition of H_2_S had no effect on the Soret band indicating that the Mn was fully reduced and was not further affected by H_2_S. Similarly, the Q band underwent a red shift from 557 nm to 562 nm after DTT addition and was unaffected by subsequent addition of H_2_S.

We also added 1 mM H_2_O_2_ to 1 μM MnTE and observed a dramatic decrease in the Soret band but no effect on the wavelength (451 mn) for up to 20 min (Figure 8C,F). Subsequent addition of 100 μM H_2_S produced a blue shift in the Soret band to 437 nm but the amplitude was greatly diminished compared to either H_2_S alone or H_2_S after DTT (Figure 8A,B). The Q band became progressively smaller and essentially disappeared after 20 min of H_2_O_2_. It did not reappear after subsequent addition of H_2_S. In agreement with published data [31,36], these data suggest that high concentrations of H_2_O_2_ degrades MnTE and the MnP becomes inactivated.

#### 3.5.3. Reversibility of MnTE-H_2_S Interaction

Because H_2_S is volatile and rapidly disappears from solution [37], we assumed that if the reaction of H_2_S with MnTE was reversible we would see a reoxidation of MnTE after all of the H_2_S was either consumed in the reaction or the excess had volatilized off. Addition of 100 μM H_2_S to 1 μM MnTE in normoxia produced an initial left shift in the Soret band from 452 nm to 437 nm; however, at 60 min this began to shift right and by 120 it returned to 451 nm and 452 nm at 150 min (Figure 9A). In addition to this, the entire spectrum shifted upward (more pronounced in the UV/blue region than the red), consistent with the additional contribution of polysulfides, which have a broad shoulder in this wavelength range. These results show that Mn is reoxidized to Mn(III) after the H_2_S is gone. The Q band underwent a rapid red shift from 558 nm to 561 nm after addition of H_2_S. At 90 min it returned to 558 nm and continued a blue-shift to 552 nm at 120 min and 548 nm at 150 min, although the peak was broadened and not as pronounced at this time.

To determine if the re-oxidation of Mn(II) was O_2_ dependent we repeated the above experiments in a closed cuvette to prevent H_2_S volatilization, but with excess O_2_ and 1 μM MnTnBuOE. We assumed that with excess O_2_ (21% O_2_, ~260 μM O_2_) relative to H_2_S (50 μM) the spectrum would initially show the characteristic H_2_S-mediated blue and red shifts of the Soret and Q bands, respectively and then these would return back to the starting, oxidized MnTnBuOE spectrum after the H_2_S was consumed. Indeed, this was observed (Figure 9B).

We then examined the effects of low H_2_S concentrations to determine if Mn was reoxidized and recycled after the H_2_S had been metabolized. Stepwise addition of 0.5 to 6 μM H_2_S to 1 μM MnTE did not affect either Soret or Q band (Figure 9C,D), suggesting that the H_2_S was quickly depleted and the MnTE re-oxidized. The Soret band began to become blue-shifted at 10 μM H_2_S as demonstrated by a blue shoulder that appeared at approximately 438 nm. The Soret band was completely shifted to 437 nm by 30 μM H_2_S and the Q band was red-shifted from 556 nm to 562 nm (Figure 9C,D). These results suggest that at low H_2_S concentrations the manganese is oxidized back to Mn(III) because of rapid depletion of H_2_S. However, as H_2_S concentration increases there is sufficient substrate to keep the manganese reduced or the polysulfide generated from the reaction maintains manganese in the reduced state.

These hypotheses were further explored by examining the reaction of 1 μM MnTE and H_2_S at smaller increments of H_2_S concentrations in both normoxia and hypoxia and with H_2_S_2_. As shown in Figure 10A–E, in normoxia 5 μM H_2_S produced a slight blue shift in the Soret peak from 452 nm to 451 nm and a red shift in Q from 556 nm to 559 nm and then to 558 nm at 20 min. As H_2_S concentrations were increased from 10 to 25 μM in normoxia the blue shift in the Soret peak became more pronounced (451, 449, 439, and 438 nm, respectively) and the tendency to return back to 452 at 20 min was decreased (450, 450, 449, and 447 nm, respectively). In addition, a blue-shifted shoulder in the t = 0 min became more pronounced as H_2_S concentration was increased from 5 to 15 μM before the complete blue shift at 20 μM H_2_S. The shoulder at t = 20 min became more pronounced as H_2_S concentration increased (and reduction of MnP becomes thermodynamically favored over its reoxidation) and the spectrum had the appearance of a split Soret band at 25 μM H_2_S. The red-shift in the Q band followed a similar pattern. Under hypoxic conditions the Soret peak was blue-shifted and the Q peak red-shifted at both 0 and 20 min irrespective of the H_2_S concentration (5–25 μM) added to MnTE because of the removal of oxygen and thus the sustained MnP reoxidation step.

As persulfides and polysulfides can be either oxidants or reductants, we then examined the effects of H_2_S_2_ on the MnTE spectrum to determine if persulfide formation from MnTE-catalyzed oxidation of H_2_S could account for the restoration in the Soret peak from the reduced Mn(II)TE (438 nm) back to the oxidized Mn(III)TE (452 nm). Increasing the H_2_S_2_ concentration from 0, 1, 3, 10, 30 μM produced a blue shift in the Soret peak from 452, 451, 450, 449 to 436 nm with the appearance of a progressively broader blue shoulder from 1 to 10 μM H_2_S_2_ (Figure 10F) and a red shift in the Q band. Thus, the persulfide can also act as a Mn reductant. This is not surprising as, contrary to intuition, oxidation of sulfide to a persulfide produces a better reductant than the original sulfide (Fukuto et al. 2018 [38]). Although we could not discount the possibility that there was some contamination of the H_2_S_2_ with H_2_S, this is unlikely to account for the spectral shift because 10 μM H_2_S_2_ produced as significant blue shift as15 μM H_2_S (Figure 10C). Collectively, these results suggest that the slow recovery (red shift) in the Soret peak is due to the dissociation of the sulfur from the manganese and subsequent reoxidation by oxygen. These results also suggest that MnPs oxidize per- and polysulfides.

## 4. Discussion

Our results show that all three SOD-mimetic MnPs, MnTE, MnTnHex, and MnTnBuOE, catalyze the polysulfide formation from Na_2_S (which immediately produces H_2_S when dissolved [37]). These reactions are both MnP and H_2_S concentration dependent and can operate at sub-micromolar MnP concentrations. Unfortunately, optical quenching of the analytic fluorophore by the MnPs prevented analysis above 3 μM MnP. Hydrogen persulfide (H_2_S_2_) appears to be the initial product while other polysulfides, H_2_S_3_-H_2_S_6_ appear later and in lesser amounts. MnP-catalyzed polysulfide production from H_2_S appears to consume oxygen and involve redox cycling employing Mn reduction by H_2_S and reoxidation by oxygen. MnTBAP was minimally efficacious, likely because its reduction potential (−194 mV E_½_ vs. NHE) is relatively far removed from SODs and the other MnPs of this study (~+300 mV; [7]) which prevented redox cycling.

### 4.1. Quantification and Identification of Polysulfides Produced by MnP Catalysis of H_2_S Oxidation

An estimate of the amount of polysulfide produced at 90 min by MnP metabolism of H_2_S can be obtained from the calibration curve of SSP4 fluorescence vs. the mixed polysulfide, K_2_S_n_ (K_2_S_n_ where *n* = 1–7), or individual polysulfides, Na_2_S_2_, Na_2_S_3_, and Na_2_S_4_ (Figure 3F). Comparison of Figure 3E,F suggests that 3 μM MnP and 300 μM H_2_S produces approximately 52 (MnTE), 21 (MnTnHex), and 31 (MnTnBuOE) μM of polysulfide. This would be equivalent to 104, 42, and 62 μM sulfur in H_2_S_2_, or 156, 63, and 93 μM sulfur in H_2_S_3_, suggesting relatively efficient catalytic activity. Figure 3F also indicates that SSP4 reacts on a molar basis with the polysulfide, not with individual sulfur molecules in the polysulfide.

Results from the mass spectrometry study show that the initial polysulfide produced from MnP catalysis is hydrogen persulfide (H_2_S_2_) as this species appears first and in the greatest amount (Figure 4). It is not known if MnPs are able to catalyze larger polysulfides directly from H_2_S, although their delayed appearance compared to H_2_S_2_ suggests that if this occurs the reaction is slower. Polysulfide formation by MnTnHex provides circumstantial evidence for this as all these reactions appear somewhat delayed compared to MnTE. Figure 5 also suggests that polysulfides with more than two sulfur atoms may be formed from further oxidation of H_2_S or H_2_S_n_ and subsequent reorganization of the polysulfides. MnTE metabolism of derivatized H_2_S_4_ (Figure 6) supports this contention; as the concentration of MnTE is increased the concentration of derivatized H_2_S and H_2_S_2_ decrease while the concentrations of H_2_S_4_, H_2_S_5_, and H_2_S_6_ all increase. H_2_S_3_ appears as the transition as its concentration did not change. Spontaneous formation of larger polysulfides is also possible and additional studies are required to resolve the ability of MnPs to metabolize polysulfides with S > 2.

### 4.2. H_2_S and MnP Redox Cycling

Previous studies have shown that the absorption spectrum of Mn(III)TE in argon-purged PBS at pH 7.4 has a Soret peak at approximately 454 nm and another small peak (Q band) at around 560 nm. When the MnTE is fully reduced with 500 μM ascorbate these shift to 438 and 565 nm, respectively [35]. These values are similar to our values, 452 and 558 nm when oxidized and 436 and 562 nm when reduced with 1 mM DTT (Figure 8B,E).

When 100 μM or 300 μM H_2_S were incubated with 1 μM MnTE in normoxia the Soret and Q bands were blue- and red-shifted to the same extent consistent with the reduction of oxidized Mn(III) to Mn(II) (Figure 8A,D). If allowed access to room air in uncovered cuvettes these peaks return to their original wavelengths (Figure 9A) suggesting that the Mn is re-oxidized to Mn(III). If the cuvettes are covered but with excess oxygen relative to H_2_S (Figure 9B) the Mn remains reduced for a longer period but eventually it also becomes reoxidized, presumably after the H_2_S is consumed. This suggests that oxygen re-oxidizes Mn(II)P. It is not clear why Mn appears to remain reduced when exposed to H_2_S even in the presence of excess oxygen, in hypoxia or normoxia, even though it likely redox cycles. A number of factors could account for this; (1) The oxidation reaction could be considerably faster than the H_2_S reduction process; (2) if the sulfur is bound to Mn and only released after the oxidant binds to the Mn, the Mn could immediately bind another sulfur and be reduced; (3) if the reaction products also reduce Mn as equilibrium is approached, this would keep Mn reduced even after the initial reactants are consumed. Any one of these options would give the appearance of a continually reduced Mn at the timescale of our spectral measurements, especially the third option as Mn is reduced by H_2_S_2_ (Figure 10F), although additional experiments are required to evaluate the first two.

The trade-off between Mn reduction and oxidation and the domination of H_2_S-mediated reduction was also reflected in the Soret peak at various H_2_S concentrations in normoxia where O_2_ concentration was around 260 μM, i.e., room air (Figure 9C,D). With 1 μM MnTE the Mn remained oxidized as H_2_S increased from 0.05 μM to 6 μM at which point there was a slight blue shoulder in the Soret band. At 10 μM H_2_S the blue shoulder became quite pronounced and the Soret peak was completely blue-shifted by 30 μM H_2_S. The oxygen dependency of these wavelength shifts was confirmed by comparing the effects of 5–25 μM H_2_S on the MnTE spectrum in normoxia and hypoxia (Figure 10) where Mn re-oxidation was only observed in normoxia.

### 4.3. Proposed Mechanism of MnP Oxidation of H_2_S

Our experiments indicate that O_2_ is consumed during MnP-catalyzed oxidation of H_2_S suggesting the following reactions. First, 2Mn(III) are reduced by two hydrosulfide anions generating two hydrosulfide radicals (Equation (1)). The hydrosulfide radicals then react with each other to produce the hydrogen persulfide (Equation (2)). The reduced Mn(II) is then re-oxidized by O_2_ forming Mn(III) and superoxide (O_2_•^−^; Equation (3)) and the latter is then spontaneously, or catalyzed by MnP, dismuted to hydrogen peroxide (H_2_O_2_) and water (Equation (4)). Hydrogen peroxide produced in Equation (4) can re-oxidize either Mn^II^P^4+^ or Mn^III^P^5+^ which results in the formation of high-valent MnP, Mn^IV^ (Equation (5)) or Mn^V^P (Equation (6)), respectively. The high-valent MnP can then oxidize HS^−^ (Equations (7) and (8), respectively) and the hydrosulfide radicals formed will also produce H_2_S_2_ (Equation (2)).
2Mn^III^P^5+^ + 2HS^−^ → 2Mn^II^P^4+^ + 2HS^•^(1)
2HS^•^ → H_2_S_2_(2)
2Mn^II^P^4+^ + 2O_2_ → 2Mn^III^P^5+^ + 2O_2_^•−^(3)
2O_2_^•−^ + 2H^+^ → H_2_O_2_ + O_2_(4)
Mn^II^P^4+^ + H_2_O_2_ → O = Mn^IV^P^4+^ + H_2_O(5)
Mn^III^P^5+^ + H_2_O_2_ → (O)_2_Mn^V^P^3+^ + 2H^+^(6)
O = Mn^IV^P^4+^ + HS^−^ + 2H^+^ ↔ Mn^III^P^5+^ + HS^•^ + H_2_O(7)
(O)_2_Mn^V^P^3+^ + HS^−^ + 2H^+^ ↔ O = Mn^IV^P^4+^ + HS^•^ + H_2_O(8)
Higher order polysulfides may be formed from the reaction of Mn(III) with HS_2_^−^ (Equation (9));
Mn^III^P^5+^ + HS_2_^−^ → Mn^II^P^4+^ + HS_2_^•^(9)
which can react with a hydrosulfide radical (Equation (10)) or another persulfide radical (Equation (11)) to produce longer chain polysulfides.
HS_2_^•^ + HS^•^ → H_2_S_3_(10)
HS_2_^•^ + HS_2_^•^ → H_2_S_4_(11)

## 5. Conclusions

It is now generally accepted that when peroxide concentration is carefully regulated it is an important signaling entity, but when in excess it can produce oxidative distress with pathophysiological consequences that can affect all organ systems [39,40,41,42,43,44,45,46,47,48,49,50,51,52,53,54,55,56]. MnPs were originally developed as SOD mimetics to counter oxidative (dis)stress but have gained recent attention through their ability to employ H_2_O_2_ (which MnPs either generate themselves or is produced during chemo- or radiotherapy) for oxidative modification of proteins involved in cellular signaling processes. It is this reaction that has been the focus of much of the therapeutic functions of MnPs. The chemical and biological similarities between ROS and RSS, as noted in the introduction, suggest that some of the effects attributed to ROS may actually be due to RSS. In the present study we have extended this uncertainty to MnPs as they appear to interact with RSS as readily as they do with ROS. While we do not mean to imply that all the actions of MnPs are mediated through RSS metabolism, clearly the extent of RSS metabolism warrants consideration.

## Figures and Tables

**Figure 1 antioxidants-08-00639-f001:**
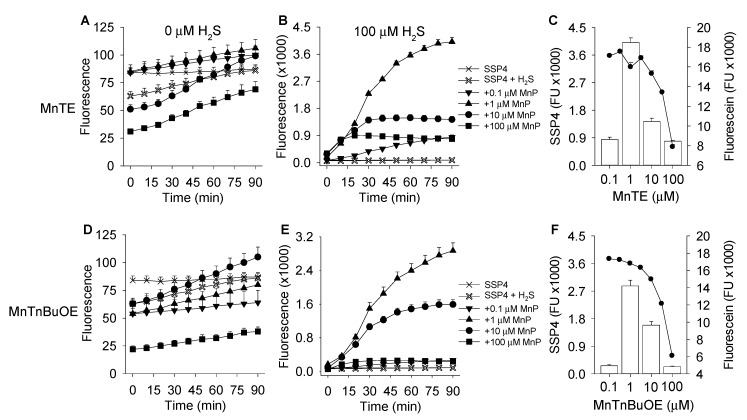
Effects of MnTE (**A**–**C**) and MnTnBuOE (**D**–**F**) on polysulfide production (SSP4 fluorescence) in the absence (0 μM H_2_S) or presence of 100 μM H_2_S in buffer. Without H_2_S there is little increase in SSP4 fluorescence, whereas in the presence of H_2_S SSP4 fluorescence initially increases with MnP concentration up to 1 μM MnP and decreases at 10 μM MnP thereafter. (**C**,**F**) Comparison of the effects of MnP concentration on SSP4 fluorescence produced by 100 μM H_2_S at 90 min (bars) to the effects of MnP on fluorescence produced by 1 μM fluorescein (filled circles). MnP concentrations at and above 10 μM progressively inhibit fluorescence irrespective of fluorophore. Mean + SE, *n* = 4 all experiments.

**Figure 2 antioxidants-08-00639-f002:**
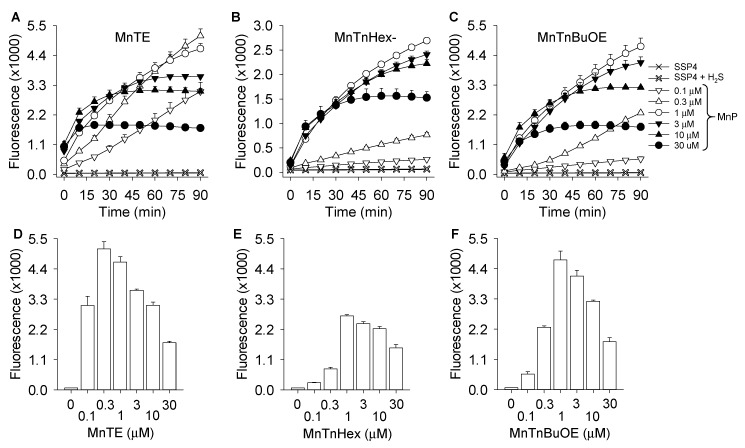
Concentration- and time-dependent effects of MnTE (**A**) and MnTnHex (**B**) and MnTnBuOE (**C**) on polysulfide production (SSP4 fluorescence) in the presence of 100 μM H_2_S and their respective 90 min averages (**D**–**F**). Experiments in buffer; mean +SE, *n* = 4 all experiments.

**Figure 3 antioxidants-08-00639-f003:**
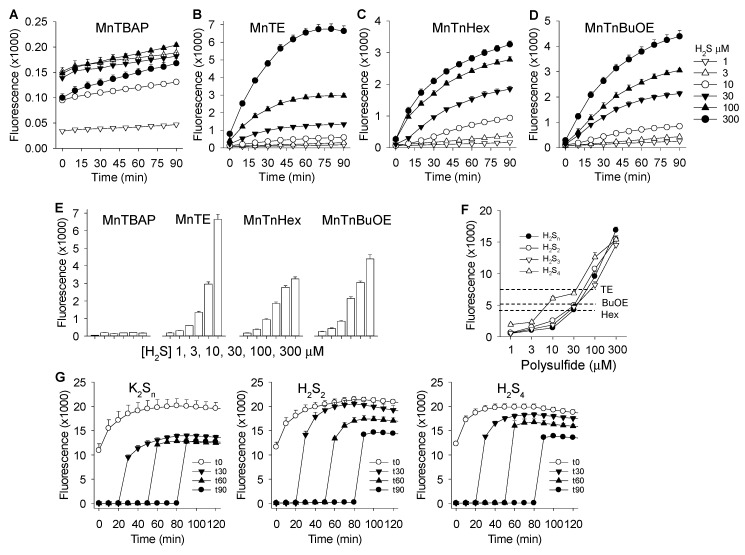
MnP (3 μM) catalyzed polysulfide generation is dependent on H_2_S concentration. (**A**) MnTBAP, (**B**) MnTE, (**C**) MnTnHex (**D**) MnTnBuOE, (**E**) average production at 90 min, all MnPs on same scale. MnTE was the most efficacious at highest H_2_S concentrations but less so at lower H_2_S concentrations; the effects of MnTBAP, which lacks SOD mimetic activity were minimal. (**F**) calibration curve (SSP4 fluorescence vs. mixed polysulfide (K_2_S_n_, *n* = 1–7), or individual polysulfides, K_2_S_2_, K_2_S_3_, and K_2_S_4_ concentration. Dashed lines indicate SSP4 fluorescence produced by MnPs from 300 μM H_2_S (values from E). (**G**) MnTE (1 μM) added to 300 μM of mixed polysulfide (K_2_S_n_, *n* = 1–7), Na_2_S_2_ or Na_2_S_4_ and incubated for 0, 30, 60, or 90 min prior to addition of SSP4 shows decreased SSP4 fluorescence with time suggesting MnTE reacts with polysulfides. All values are mean + SE *n* = 4.

**Figure 4 antioxidants-08-00639-f004:**
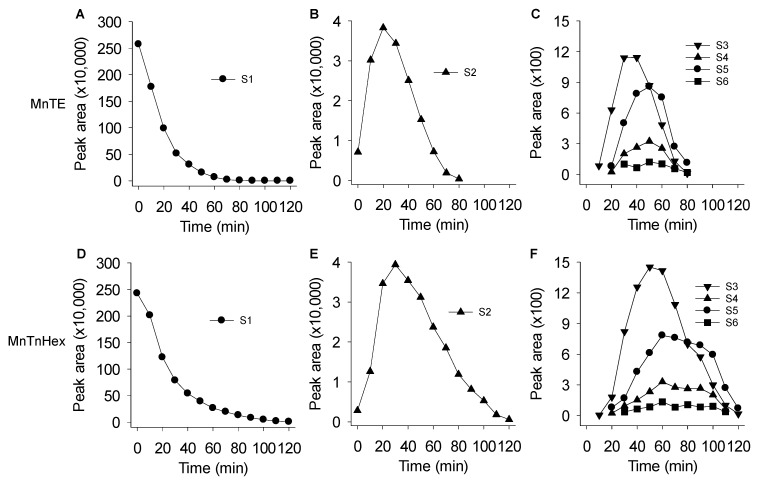
Mass spectrometric detection of H_2_S and polysulfides produced by reaction of 1 mM Na_2_S with 1 μM MnTE (**A**–**C**) or 1 μM MnTnHex (**D**–**F**). MnTE and MnTnHex produced an exponential decrease in H_2_S (**A**,**D**), and initially produced a transient increase in H_2_S_2_ (**B**,**E**) followed by (in decreasing amounts) H_2_S_3_, H_2_S_5_, H_2_S_4_, and H_2_S_6_ (**C**,**F)**. All values are means of two replicates.

**Figure 5 antioxidants-08-00639-f005:**
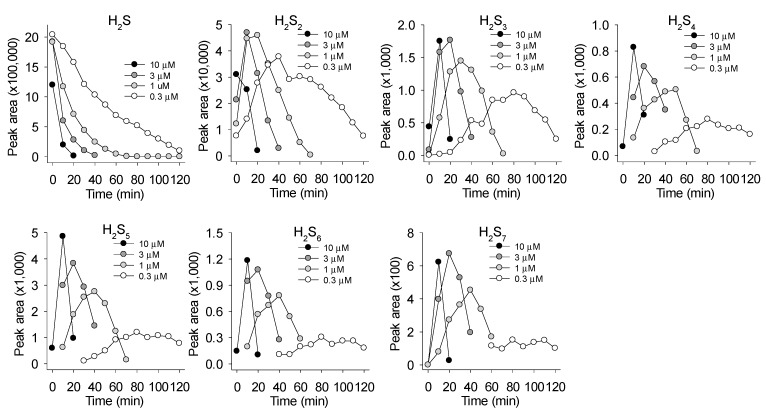
Mass spectrometric detection of H_2_S and polysulfides produced by reaction of 1 mM Na_2_S with 0.3, 1, 3, and 10 μM MnTE. As the concentration of MnTE increased the rate of H_2_S consumption and polysulfide production as well as polysulfide consumption increased. All values are means of two replicates.

**Figure 6 antioxidants-08-00639-f006:**
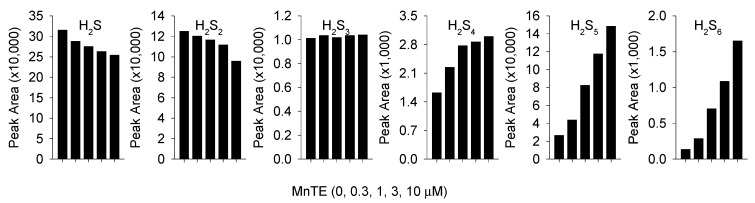
Mass spectrometry detection of H_2_S and polysulfides produced by reaction of 1 mM Na_2_S_4_ with 0.3, 1, 3, and 10 μM MnTE 30 min after reaction products were derivatized with 100 mM iodoacetate (IAM). MnTE appears to react with the IAM derivatized polysulfides, consuming H_2_S and H_2_S_2_ and producing H_2_S_4_, H_2_S_5_, and H_2_S_6_. Control (MnTE = 0 μM) values are *n* = 8, all others are average of two replicates.

**Figure 7 antioxidants-08-00639-f007:**
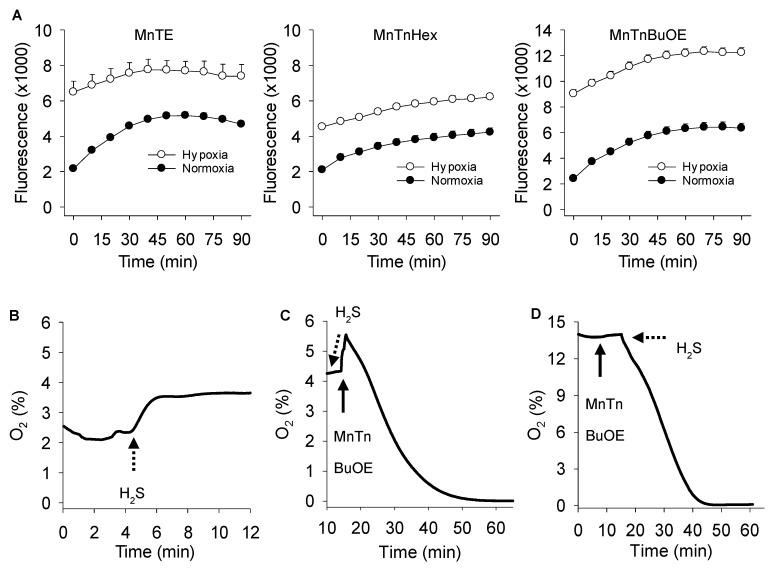
Oxygen dependency of MnP-catalyzed H_2_S oxidation. (**A**) MnP (3 μM) catalyzed polysulfide generation (SSP4 fluorescence) from 300 μM Na_2_S in normoxia or hypoxia. Normoxia panels, SSP4 fluorescence from 0 to 90 min in 21% O_2_; hypoxia panels, 90 min in 100% N_2_ then transfer to normoxia for 0–90 min, 0 min in these panels reflects 90 min prior hypoxia exposure. Polysulfide production in hypoxia is equal to, or greater than that in normoxia. All values mean + SE *n* = 4. (**B**–**D**) Traces showing direct measurement of O_2_ consumption with a fiberoptic probe. Buffer was partially deoxygenated with N_2_ and placed with the probe in a parafilm^®^ sealed vial and O_2_ monitored after addition of (**B**) Na_2_S (300 μM), (**C**) 300 μM Na_2_S then 1 μM MnTnBuOE, or (**D**) at a higher starting O_2_ with 1 μM MnTnBuOE followed by 1 mM Na_2_S. Neither Na_2_S nor MnTnBuOE alone consumed O_2_, whereas in combination all the O_2_ was consumed.

**Figure 8 antioxidants-08-00639-f008:**
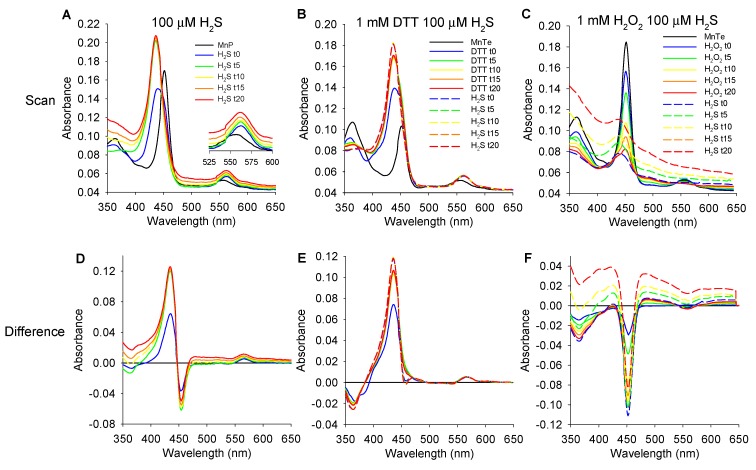
Absorption and difference spectra at 5 min intervals of 1 μM MnTE after addition of 100 μM H_2_S (**A**,**D**), 1 mM DTT for 20 min then H_2_S (**B**,**E**), or after addition of 1 mM H_2_O_2_ for 20 min then H_2_S (**C**,**F**). H_2_S and DTT produce a similar blue-shift in the MnTE Soret band indicative of reducing Mn(III) to Mn(II) and a red-shift in the Q band. A slight shoulder is visible to the right in both H_2_S and DTT Soret bands immediately after they are added (t = 0 min). H_2_O_2_ decreases the Soret band amplitude without affecting peak wavelength.

**Figure 9 antioxidants-08-00639-f009:**
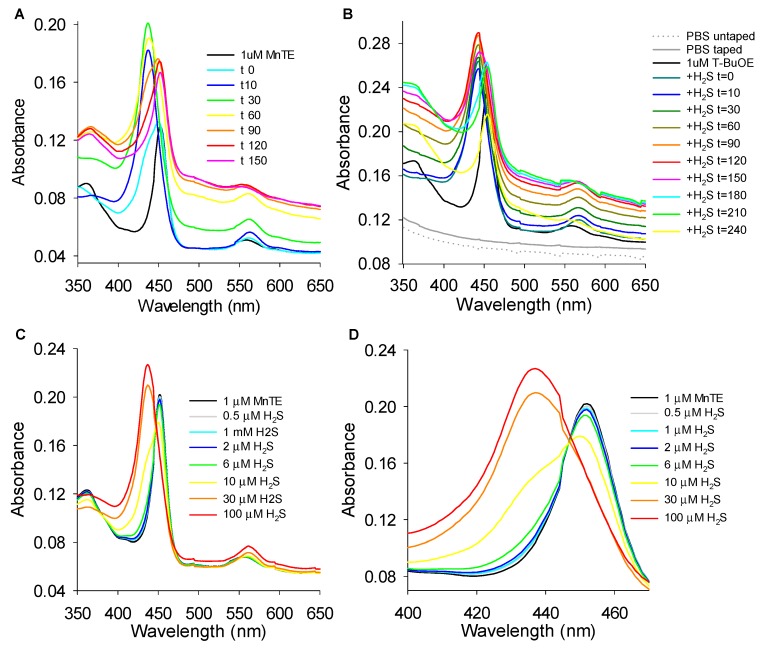
Absorption spectra of H_2_S reaction with 1 μM MnTE with or without O_2_. (**A**) In an open cuvette in normoxic buffer, addition of 100 μM H_2_S produced characteristic blue shift of the Soret band and red shift of Q band. With the open cuvette the H_2_S is lost through oxidation and volatility and after 60 min the spectrum begins to return to pre-H_2_S conditions and the maxima return to their initial values at 120 min. (**B**) In a covered cuvette with 1 μM MnTnBuOE and an excess of oxygen (~260 μM) relative to H_2_S (50 μM) the Soret and Q bands of the oxidized MnTnBuOE (black line) are blue and red shifted after addition of H_2_S and remain so for approximately 180 min until they begin to return to the spectrum characteristic of oxidized MnTnBuOE. Dotted and solid gray lines show spectra of buffer in uncovered (untaped) and covered (taped) cuvettes, respectively. Tape did not affect the MnTnBuOE spectrum. (**C**, expanded in **D**) Cumulative addition of low H_2_S concentrations (total indicated in legend) to 1 μM MnTE. H_2_S below 0.6 μM does not affect peak wavelength of either Soret or Q band, at 10 μM H_2_S a shoulder appears in the Soret band at approximately 437 nm and at 30 μM and above the Soret peak is at 437 nm. The Q band is red-shifted from 557 at 6 μM H_2_S to 560 nm at 10 μM and 562 at 30 and 100 μM H_2_S.

**Figure 10 antioxidants-08-00639-f010:**
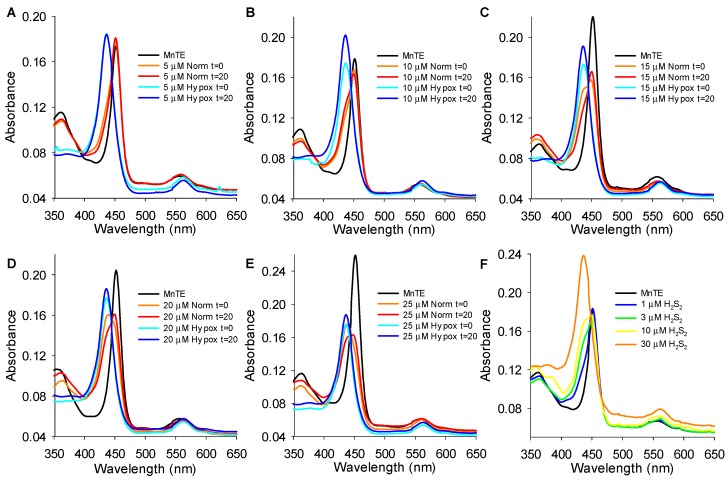
(**A**–**E**) Effects of various H_2_S concentrations on the MnTE absorbance spectrum immediately (t = 0) and 20 min (t = 20) after addition to 1 μM MnTE. H_2_S was added to MnTE in both normoxia and hypoxia. (**F**) Absorbance spectrum of 1 μM MnTE and increasing concentrations of H_2_S_2_. As H_2_S_2_ concentration increased from 3 to 10 μM, MnTE became more reduced (blue shifted) and was fully reduced at 30 μM H_2_S_2_.

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
