# Peer review of "Manganese Porphyrin-Based SOD Mimetics Produce Polysulfides from Hydrogen Sulfide"

_antioxidants, 2019, doi:10.3390/antiox8120639_

Round 1

Reviewer 1 Report

Previous studies by the authors have shown that superoxide dismutase (SOD) can oxidize sulfide to polysulfides. 

In this study the authors examine the ability of SOD mimetics to also yield  polysulfides.  

This is a very thorough study: the authors first check their fluorescent probes for artifactual reporting of polysulfide (PS) production by demonstrating that at >5 uM SOD mimetic there is mimetic dependent fluorescence-quenching of the probes.  The rest of the studies are performed at non-interfering mimetic levels.

The authors used a combination of PS probes and MS to demonstrate that the SOD-mimetics act as catalysts for PS. Furthermore, the derivatized H2S4 data indicate that PS formation does not require SOD-mimetic.  They don't come out and say this specifically in the document but should.  

The study also includes a comprehensive UV/vis spectral characterization of the SOD-mimetics during H2S catalysis. These studies, elegantly demonstrate the O2-dependance of the process and contribute to the postulated redox mechanism.

Finally, this study strengthens the evidence that the observed therapeutic benefits of SOD-mimetics may be attributed to PS as well as ROS chemistry (some might want to call metabolism).

The only criticism of this excellent study is the poor figures: Figures >5 are better colour really helps distinguish the various conditions employed.  Figs 1 and 2 the symbols and the text are barely legible as they are too small  The gray symbols and lines in Figs 1-5 should be changed (colorize).  The figure legends are different font size to avoid ransom-look add legends to text instead cut paste from fig containing the legend. 

Author Response

Reviewer 1:
Previous studies by the authors have shown that superoxide dismutase (SOD) can oxidize sulfide to polysulfides.
In this study the authors examine the ability of SOD mimetics to also yield polysulfides.
This is a very thorough study: the authors first check their fluorescent probes for artifactual reporting of polysulfide (PS) production by demonstrating that at >5 uM SOD mimetic there is mimetic dependent fluorescence-quenching of the probes. The rest of the studies are performed at non-interfering mimetic levels.

The authors used a combination of PS probes and MS to demonstrate that the SOD-mimetics act as catalysts for PS. Furthermore, the derivatized H2S4 data indicate that PS formation does not require SOD-mimetic. They don't come out and say this specifically in the document but should.
# This point has been added to the text, good suggestion!

The study also includes a comprehensive UV/vis spectral characterization of the SOD-mimetics during H2S catalysis. These studies, elegantly demonstrate the O2-dependance of the process and contribute to the postulated redox mechanism.

# Thank you!

Finally, this study strengthens the evidence that the observed therapeutic benefits of SOD-mimetics may be attributed to PS as well as ROS chemistry (some might want to call metabolism).

The only criticism of this excellent study is the poor figures: Figures >5 are better colour really helps distinguish the various conditions employed. Figs 1 and 2 the symbols and the text are barely legible as they are too small The gray symbols and lines in Figs 1-5 should be changed (colorize). The figure legends are different font size to avoid ransom-look add legends to text instead cut paste from fig containing the legend.

# The figures have been redone with larger symbols and the line thickness has been increased in the spectra for clarity. New figures are inserted directly into the text and they are much clearer. Figure legends are also now in the text with 1.15 line spacing to distinguish them from the other text. My apologies for the figures. I am not that familiar with MS Word and had difficulty getting this to work.

Reviewer 2 Report

This is a well thought out and well presented study which builds on previous work. It is well written and has a good body of evidence to support the work.

I found a couple of typos (e.g., line 391), but little else.

My only comment would be that in the early figures it says how many repeats were done, but this seems to be missing from later in the paper. I assume the spectra are representative data, but it would be good to know that this was repeated and how many times. I think this applies to Figure 7-10.

Author Response

Reviewer 2:

This is a well thought out and well presented study which builds on previous work. It is well written and has a good body of evidence to support the work.

#Thank you!

I found a couple of typos (e.g., line 391), but little else.

My only comment would be that in the early figures it says how many repeats were done, but this seems to be missing from later in the paper. I assume the spectra are representative data, but it would be good to know that this was repeated and how many times. I think this applies to Figure 7-10.

# The typos have been corrected. Thank you.

# The figures have been redone with larger symbols and the line thickness has been increased in the spectra for clarity. New figures are inserted directly into the text and they are much clearer. Figure legends are also now in the text with 1.15 line spacing to distinguish them from the other text. My apologies for the figures. I am not that familiar with MS Word and had difficulty getting this to work.

# The number of replicates in figures 7-10 (three) has been added to the Methods.